# Relieving Bottlenecks during Evacuations Using IoT Devices and Agent-Based Simulation

Moongi Choi [1], Sung-Jin Cho [2] and Chul Sue Hwang [3,*]

1 Department of Geography, University of Utah, Salt Lake City, UT 84112, USA; u1316663@utah.edu
2 Marine Research Division, Korea Maritime Institute, Busan 49111, Korea; sjcho@kmi.re.kr
3 Department of Geography, Kyung Hee University, Seoul 02447, Korea
* Correspondence: hcs@khu.ac.kr; Tel.: +82-2-961-9313

**Abstract:** Most of the existing studies on relieving bottlenecks have aimed to develop route-finding algorithms that consider structural factors such as passages and stairs, as well as human factors such as density and speed. However, the methods in providing evacuation routes are as important as the route-making algorithms because a secondary bottleneck could occur continuously during evacuations. Even if an evacuation system provides the same routes to all evacuees regardless of their locations, secondary bottlenecks could happen following the initial bottlenecks due to people rushing toward uncrowded exits all together. To address this issue, we developed a location-based service (LBS) evacuation system prototype that provides optimized-alternative routes to evacuees in real time considering their locations in indoor space. The system was designed to relieve continuous bottlenecks, which relies on installed IoT sensors and beacon machines which detect bottlenecks and provide updated routes, separately. Next, we conducted agent-based simulations to measure the system's effectiveness (evacuation time reduction and dispersion of evacuees) by changing the system parameters. Simulation results show the evacuation time decreased from 100 to 65 s, and the number of people who took a detour to avoid bottlenecks increased by 28.66% out of the total evacuees with this system. Since this system provides the theoretical solution for distributing evacuees, it can be flexibly employed to a disaster situation in a large and complex indoor environment by applying to other evacuation systems. Moreover, by adjusting parameters, we can derive maximum evacuation effectiveness in other indoor spaces. Future work will consider demographic features of population and multilayer building structure to draw a more accurate pedestrian flow.

**Keywords:** evacuation; IoT; location-based service; agent based simulation; bottleneck effect

## 1. Introduction

In modern society, the size and structure of indoor spaces are becoming larger and more complex, and this can be a major obstacle in preventing rapid evacuation in disaster situations such as fire. It is difficult to predict continuous changes in evacuation flow in complex indoor spaces including multiple exits and routes, which causes inefficiency of the evacuation process. Particularly, bottlenecks, a specific situation during an evacuation where many people rush to the same exit or a narrow space and cannot escape through the exit efficiently [1], have been a major reason for the increase in evacuation time, as well as serious injuries or casualties.

There have been two kinds of research relieving bottlenecks in an indoor space. First, research focusing on human movement has tried to find out what factors have an influence on the pedestrian flow in a narrow space. These factors include both human and structural features. They captured the change of bottleneck patterns and evacuation time responding to the number of groups in population [2], over-crowded space [3], laying obstacles in front of an exit [1,4], adjusting exit width [5], and stair width [6]. Second, there have been studies in models or algorithms to provide evacuation routes to relieve bottlenecks. Most

of the research in this field employed WSN (wireless sensor network) to make routes by collecting fire, smoke, or pedestrian locations to provide evacuation routes [7–9]. Some of the research made their own route generation algorithm based on a danger index [10] and horizontal-vertical tiering [11].

Those two areas of research, however, still have some limitations in preventing or relieving continuous bottlenecks occurring in complex spaces. They cannot always locate obstacles, because it is not possible to know the exact population or the number of groups in real time in a building. Moreover, if the indoor space consists of a complex structure, it is extremely hard to curb the continuous bottleneck occurrence because we cannot control everyone's path. Therefore, the fundamental solution of relieving bottlenecks can only be realized by addressing two challenging issues as follows:

(1)　If a system gives the same alternative routes regardless of their locations, evacuees could all rush to the same another exit, and it would end up being crowded again ($n^{th}$ bottlenecks). Thus, alternative routes should be provided instantly based on people's locations in real time to avoid continuous bottlenecks.

(2)　Evacuees might choose inefficient detour routes and eventually take a long way despite the possibility that a bottleneck in front of them might disappear soon. Therefore, the system should provide alternative routes for the evacuees who have not yet reached the bottlenecks, while keeping the original routes for the people who are already trapped in the bottlenecks to avoid collision.

To address those issues, we employed an LBS (location-based service) approach to give different safe routes to people considering their locations in real time. We aimed at providing a theoretical solution for reducing continuous bottlenecks by an LBS based prototype system, rather than establishing the system that enables its use in practice directly. For this, we adopted beacon machines, which provide the routes to personal devices over a long distance. Since the beacon machines are installed along the corridors, evacuees can receive updated routes continuously in real time as they move, helping them to avoid bottlenecks before they encounter them.

Next, we employed ABM (agent-based model) to measure the improvement of evacuation effectiveness by adjusting two system parameters (recognition time of bottlenecks and radius of beacons) and population factors (total population and population ratio complying with the alternative routes).

## 2. Materials and Methods

### 2.1. System Concept and Framework

Figure 1 shows each step of the evacuation system prototype: (1) IoT (Internet of Things) devices for sensing; (2) main system to produce evacuation routes; and (3) the route-providing system. Table 1 shows the hardware and software used in the system. In the first step, an Arduino Uno R3 with a fire detection sensor, temperature/humidity sensor, and distance sensor monitors the indoor situation. Next, we attached a ZigBee module onto the Arduino Uno R3 to transmit sensor data to the main system. ZigBee is a wireless network standard with low power consumption and low latency [12]. Sensor data were captured every second, and the main system uses multi-threading to receive the data set from each Arduino device at the same time.

Second, in the main system, maps illustrating the shortest paths were produced when a fire or bottlenecks occurred. The shortest path from each node to an exit is calculated based on the Dijkstra algorithm, which can solve the SSSP (single source shortest path) problem calculating the shortest path from a given node to the last node [13]. All maps were converted into HTML format and stored on an Apache web server.

Figure 2 shows the nodes, edges, and exits in our simulation area and Figure 3 shows the method used to recognize bottlenecks in the main system. When fire emerges, the distance between each wall starts to be measured every second by two distance sensors. During an evacuation, this distance is shorter than normal since the space between the walls is crowded. Therefore, if the shorter distance is maintained, the main system starts

to recognize it as a bottleneck. We defined this distance from each sensor as 50 cm and denoted the time parameter as a RToB (recognition response time of bottlenecks).

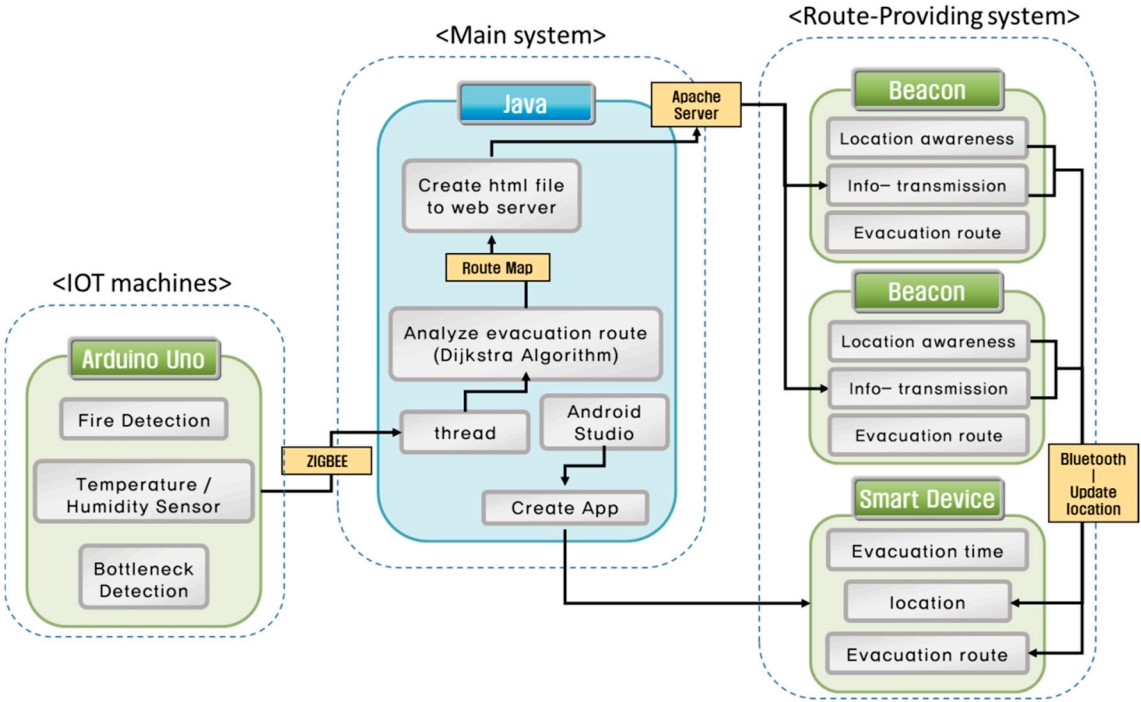

**Figure 1.** System framework.

**Table 1.** System components.

| | IoT Machines | Main System | Route-Providing System |
|---|---|---|---|
| Hardware | Arduino Uno R3 | Main server (Ram: 16 GB, Disk: SSD 256 GB, Windows 10) | RECO Beacon Smart device (Android 6.0.1) |
| | Fire detection sensor | | |
| | Temperature-humidity sensor (DHT-11) | | |
| | Distance sensing sensor (HC-SR04) | | |
| Communication | Zigbee | | |
| | Apache 2.4 Web server | | |
| Software | Arduino 1.8.1 DIGI XCTU 6.3.5 | Eclipse Neon 4.6.2 (JDK 1.8.0) ArcMap 10.1 | RECO configuration Tool 4.0.6 Android Studio 2.3.3 |

The third step begins during an evacuation. In this step, the route-providing system provides every evacuee with the closest path to the exit considering their locations through personal smart devices. Providing routes through smart device in indoor environment has been conducted in previous studies to offer route information directly to individuals [14–17]. Detour routes continuously change depending on whether bottlenecks occur or disappear nearby them; thus, evacuees can receive updated detour routes in real time when they pass through the radius of each beacon installed along the corridors. To materialize this procedure, we built a mobile application in Android Studio (ver. 2.3.3), and we assumed that all evacuees installed this application. Since the routes are provided through the individual device, there could be an issue of battery or incompatibility. However, beacon transmits the information to specific range as an area so that people who do not receive the

route information can move with others. Moreover, arrow-shaped light signals on the floor in range of the beacons could be another solution. In this system, we used individual mobile device since the main customers of the movie theater are likely to be young generation who mostly have the smart device and incompatibility issues are being disappeared in these days. Figure 4 shows a screenshot of the application. All evacuees can recognize their location, fire, whether their devices are connected to the beacons, and total evacuation time in this application. The location of evacuees was superseded by each beacon's location to reduce processing time.

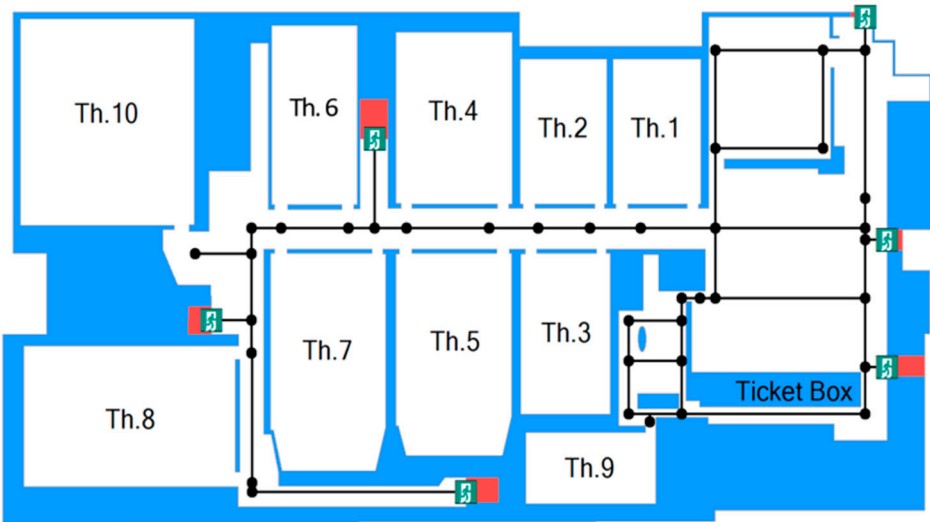

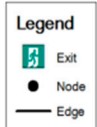

**Figure 2.** Nodes, edges, and exists in an example multiplex.

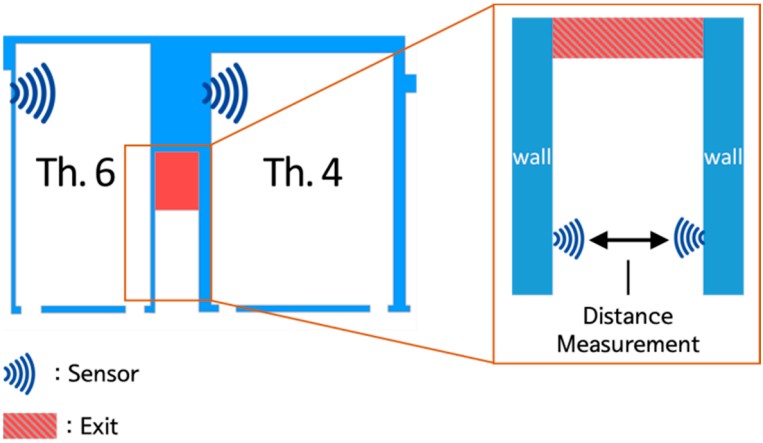

**Figure 3.** Method for recognizing bottlenecks in the main system.

Next, we conducted several tests with different scenarios to verify the route-providing system. The first scenario considered whether the routes are changed in real time by the emergence of bottlenecks. The second scenario was to confirm whether one evacuee could receive changing detour routes when passing through the radius of each beacon. The results show that all functions worked without any computational issues as Figure 4 demonstrates. (B) and (D) each show changed detour routes from (A) and (C) separately when the bottleneck occurs.

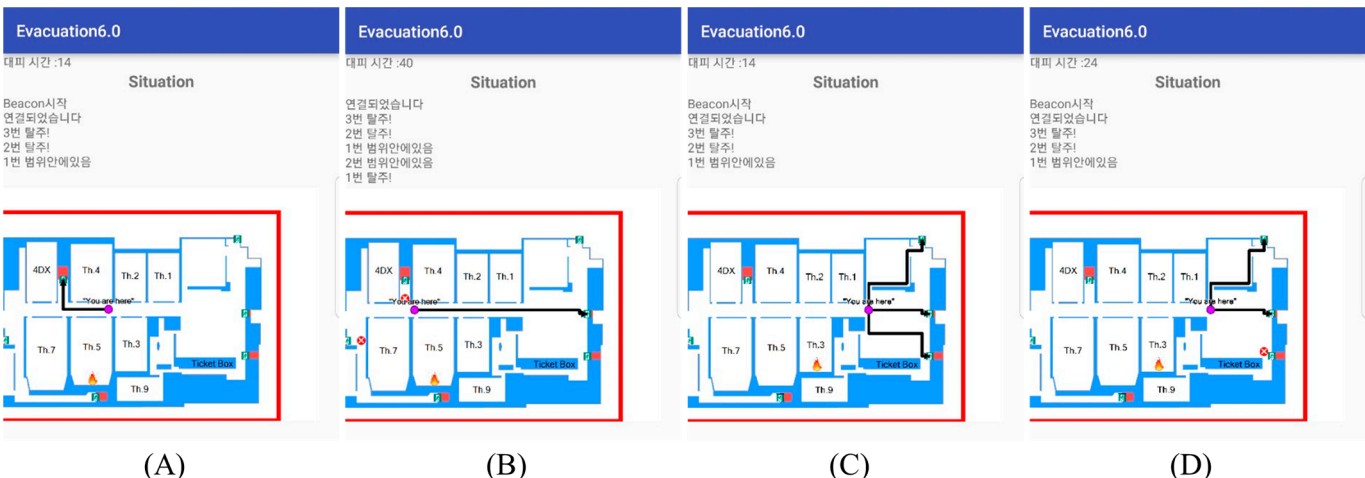

**Figure 4.** Evacuation application screen.

## 2.2. Validation Procedure by ABM Simulation

There have been two kinds of crowd simulations for evacuations, namely, multi-agent-oriented and physics-oriented simulation. The evacuation simulation technique based on multi-agents is implemented by using agent-based simulations [18–20] and a cellular automata model [2,3,21]. In this model, the dynamics of population mobility are shown as agents moving or cell values changed by interaction among themselves. On the other hand, physics-oriented simulations have included the use of a social force model [22–24], fluid dynamics model [25–27], gas kinetic model [28,29], and lattice gas model [30–32], which are based on various physics laws of dynamics that can be used to calculate the flow of pedestrian movement.

Among these simulation methods, an ABM has advantages of handling each agent's choice [33], defining characteristics of each agent [20], and illustrating interactions with the surrounding environment [34]. Since the interaction among agents is the key factors in determining movement patterns in this study, and evacuees make their own decisions regarding their movements according to the routes provided, it is appropriate to regard them as agents. Moreover, the ABM would be extremely complex, so that the ODD protocol (overview, design concepts, and details) is needed to make model descriptions more understandable and less subject to criticism [35]. The ODD describes each process and design of the model, which consists of 7 categories (purpose; entities, state variables and scales; initialization; input data; sub-models) [36].

### 2.2.1. ODD of Simulation for Environment Initialization

1.     Purpose

The purpose of this simulation is to calibrate speed and shoulder width parameters. Through this process, we can establish the initial simulation environment to validate the evacuation effectiveness of the system.

2.     Entities, State variables and Scale

The Entities are evacuees that are created individually as agents when the simulation starts and split into theaters according to the seating capacity of each. All agents have their shoulder width and speed parameters as state variables that are calibrated by multiple simulations. As for the scale, we built the virtual multiplex space that consists of 10 theaters (Th. 1 to 10) and has six main exits. The total seating capacity is 1846, excluding public spaces, as shown in Figure 5. Length and width are 110 m, 60 m separately.

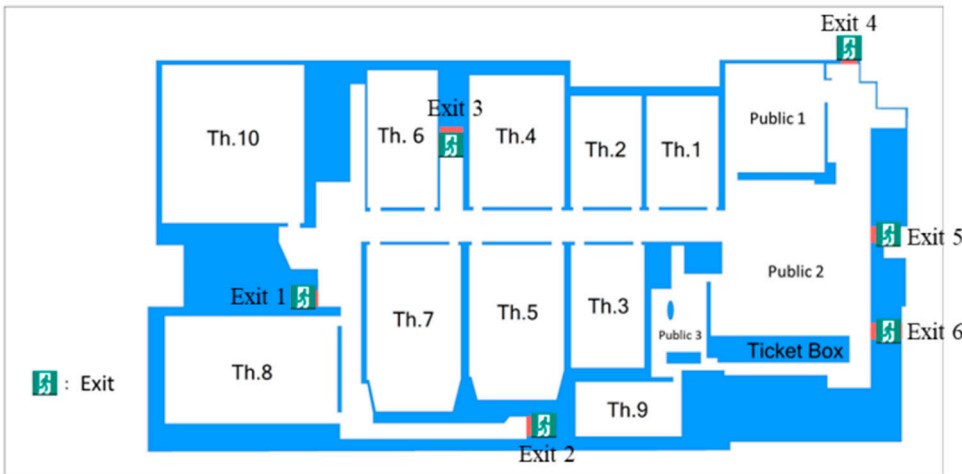

**Figure 5.** Simulation background.

3. Scheduling and Model Design

As most multiplexes announce the shortest path from each theater to the closest exit, all agents move to the closest exit from their initial locations when the evacuation begins. The speed of agents decreases once they encounter bottlenecks. If all evacuees succeed in evacuation, the schedule of simulation terminates. We implemented 1750 evacuation simulations to calibrate the speed and shoulder width variables, assuming that there is no evacuation system (5 speeds × 7 shoulder widths × 50 repetitions). The moving speed ranges were 1.2–1.8, 1.3–1.9, 1.4–2.0, 1.5–2.1, and 1.6–2.2 m/s (triangular distribution) [5,18], and shoulder widths ranged from 47 to 53 cm in 1 cm increments [37]. From every simulation, we extracted the density in front of the exit; the number of evacuees who passed through the exit; and total evacuation time in each simulation to compare pairs of speeds and shoulder widths with the experiment results from Liddle et al. [38]. Evacuee density was collected per meter squared.

4. Initialization, Input data, and Sub model

We initially set the occupancy of the multiplex to 85% capacity, i.e., 1598 people in the theaters and 445 people in the public spaces. All people sat at regular intervals in each theater and were randomly placed in the public spaces at the early stage. There were no input data and sub model in this model.

2.2.2. ODD of the System-Introduced Simulation

1. Purpose

The purposes of the system-introduced simulation are: (1) to evaluate the evacuation effectiveness (the reduction time in evacuation time and dispersion of population) when the system is introduced; (2) to acquire the best evacuation effectiveness of the system through adjusting parameters (RToB and beacon radius) and the sensitivity analysis (by total population and the number of people who follow the alternative routes).

2. Entities, State variables and Scale

The entities are evacuees as agents. They move 1.2 to 1.8 m/s as an initial speed and have 51 cm shoulder width. State variables are the RToB and the beacon radius, playing a role as factors in finding the best evacuation effectiveness by multiple simulations. For sensitivity analysis, total population, and a total compliance rate of the provided alternative route are set to the state variables.

3. Scheduling and Model design

In the case of a fire, agents move to the closest exit. In the meantime, the beacons provide evacuees with alternative route maps when they pass through each beacon's radius.

Beacons were assumed to be installed in front of each exit, as shown in Figure 6. If the exit where the agents in the beacon radius are expected to reach was crowded, the detour routes provided were changed to avoid bottlenecks. To measure the evacuation effectiveness and optimize the system settings, we conducted multiple executions varying the beacon radius (2 to 5 m in 0.5 m increments) and RToB (3 to 13 s in 2 s increments). The total evacuation time and the number of people taking a detour were collected during each simulation. We set the maximum beacon radius to 5 m to avoid overlaps among beacons.

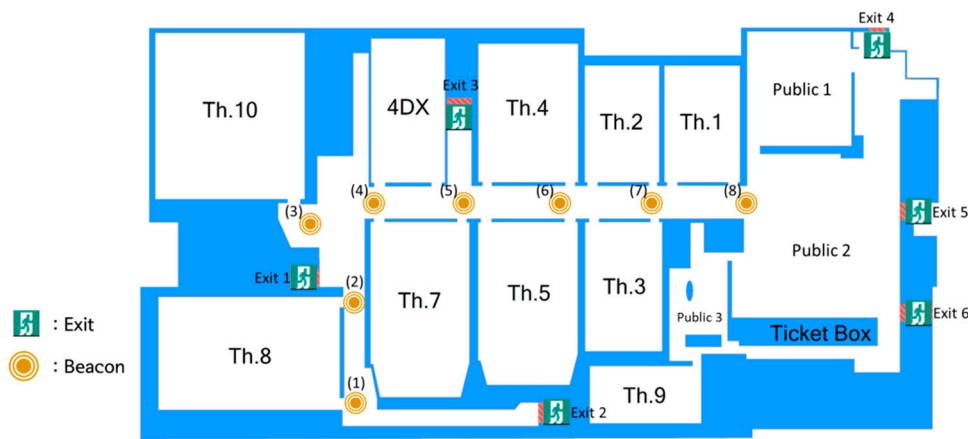

**Figure 6.** Simulation background (beacon locations).

4. Initialization, Input data, and Sub model

Initial setting of shoulder width and agents' speed are set following the result of the "simulation for environment initialization". There are no input data nor sub model.

## 3. Results

### 3.1. Result of the Simulation for Environemt Initialization

Figure 7 shows the density of evacuees in front of exit 1 and 3 by various shoulder widths (x-axis) and speeds (y-axis). According to the Liddle et al. (2009) [38], the density in front of an exit was $4.79 \pm 1.00$ people/$m^2$ and $3.52 \pm 0.87$ people/$m^2$ when the exit width was 1.6 m and 2.5 m, respectively. Since all exit widths were 2 m, we removed pairs that do not range from 2.65 to 5.79 people/$m^2$. As a result, the pairs (1.2 to 1.8 m/s, 54 cm and 1.2 to 1.8 m/s, 53 cm) were removed.

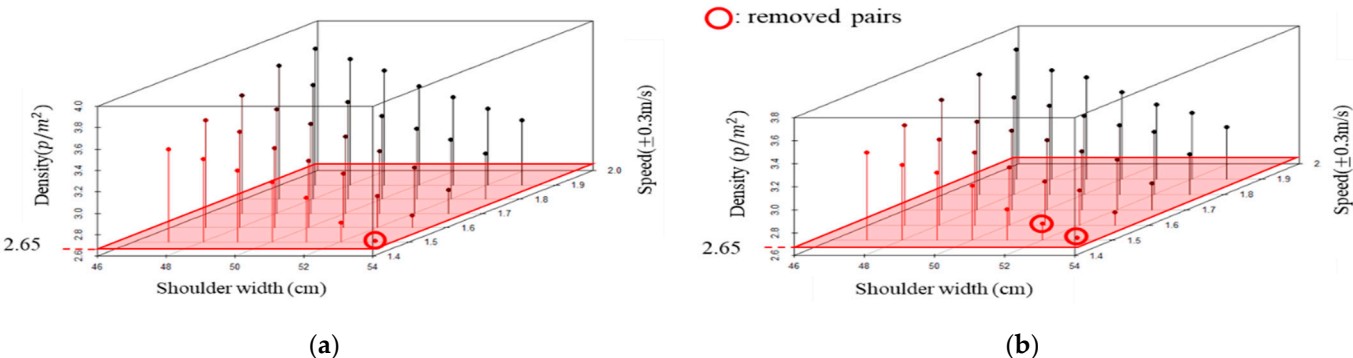

(**a**)  (**b**)

**Figure 7.** Population density results: (**a**) exit number 1; (**b**) exit number 3.

Table 2 shows the total number of evacuees who passed through an exit each second; this result was compared with the results from Liddle et al. (2009) [38] using Spearman's rank-order correlation. In their experiment, evacuees are positioned in front of an exit before starting evacuation. In the simulation we conducted, however, all evacuees start

to move far from the exit; thus, the time when the bottleneck starts to occur is later than in Liddle et al. (2009). To ensure that our experiment was consistent, we collected the number of people 50 s before the evacuation was completed ("E" in Table 2). Table 3 shows the correlation analysis result. Since the 1.2 to 1.8 m/s and 51 cm pair showed the largest correlation coefficient with a 99.9% confidence level, we used this pair as fixed parameters in the system-introduced simulation.

**Table 2.** Total number of people who pass through the exit per second.

| Time (Seconds) | No. of Evacuees (Liddle et al., 2009) | Time (E: Finish Time of Evacuation) | Average Number of Evacuees at 1.3rd Exit | | | | |
|---|---|---|---|---|---|---|---|
| | | | 1.2 to 1.8 m/s 47 cm | 1.2 to 1.8 m/s 48 cm | ... | 1.6 to 2.4 m/s 52 cm | 1.6 to 2.4 m/s 53 cm |
| 5 | 30 | E − 45 | 12.6 | 12.6 | ... | 14.8 | 13.6 |
| 10 | 25 | E − 40 | 12.6 | 12.4 | ... | 13.8 | 15.4 |
| 15 | 22 | E − 35 | 13.4 | 12.2 | ... | 14 | 13.8 |
| 20 | 22 | E − 30 | 11.6 | 12.6 | ... | 13.8 | 14.6 |
| 25 | 17 | E − 25 | 12.4 | 13.2 | ... | 14.2 | 13.8 |
| 30 | 17 | E − 20 | 11.8 | 12.2 | ... | 13.2 | 14.2 |
| 35 | 12 | E − 15 | 12.4 | 12 | ... | 14.6 | 13 |
| 40 | 15 | E − 10 | 12.0 | 12 | ... | 13.4 | 14.2 |
| 45 | 11 | E − 5 | 12.8 | 12.2 | ... | 15.2 | 15.2 |
| 50 | 10 | E | 7.4 | 5.8 | ... | 6.8 | 6.4 |

**Table 3.** Correlation analysis results.

| | 1.2 to 1.8 m/s 48 cm | 1.2 to 1.8 m/s 49 cm | 1.2 to 1.8 m/s 51 cm | 1.3 to 1.9 m/s 50 cm | 1.3 to 1.9 m/s 51 cm | 1.3 to 1.9 m/s 52 cm | 1.5 to 2.1 m/s 52 cm | 1.5 to 2.1 m/s 53 cm | ... |
|---|---|---|---|---|---|---|---|---|---|
| Correlation coefficient | 0.717 * | 0.683 * | 0.858 ** | 0.785 ** | 0.650 * | 0.719 * | 0.650 * | 0.737 * | ... |
| Significance level (2-tailed) | 0.019 | 0.029 | 0.001 | 0.007 | 0.042 | 0.019 | 0.042 | 0.015 | ... |
| Total number of compared pairs | 10 | 10 | 10 | 10 | 10 | 10 | 10 | 10 | ... |

* Confidence level: 95% (2-tailed), ** confidence level 99% (2-tailed).

### 3.2. Result of System-Introduced Simulation

Figures 8 and 9 show the simulation results when there is no system and when there is one, respectively. In Figure 8, agents did not receive the detour route information; thus they moved to the closest exit from their initial location until the simulation ends. Bottlenecks took place at exits 1, 3, 5 and last long, although other exits were available. The average total evacuation time resulted in approximately 375 s.

In Figure 9, all agents moved to the closest exit 20 s after the evacuation began. After 60 s, however, people who were located within the radius of the 6th beacon took a detour to avoid the bottleneck on exit 3. Similarly, people who were within the radius of the 2nd beacon changed their routes to exit 2, even though exit 1 is closer. People who passed through the 8th beacon chose the exits between 4, 5, or 6 evading bottlenecks throughout the simulation. The result showed that total evacuation time decreased by 70 to 100 s compared to the non-system situation.

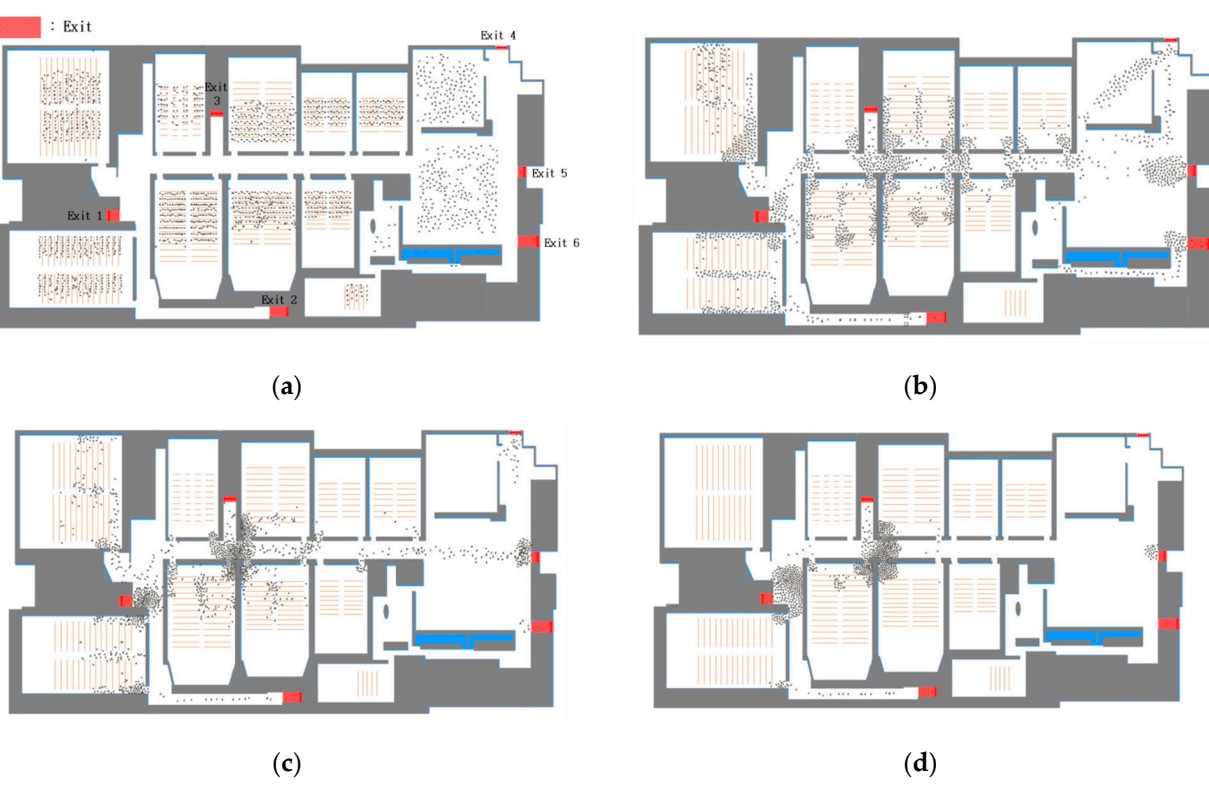

**Figure 8.** Simulation results without the evacuation system: (**a**) evacuation start; (**b**) after 60 s; (**c**) after 120 s; (**d**) after 240 s.

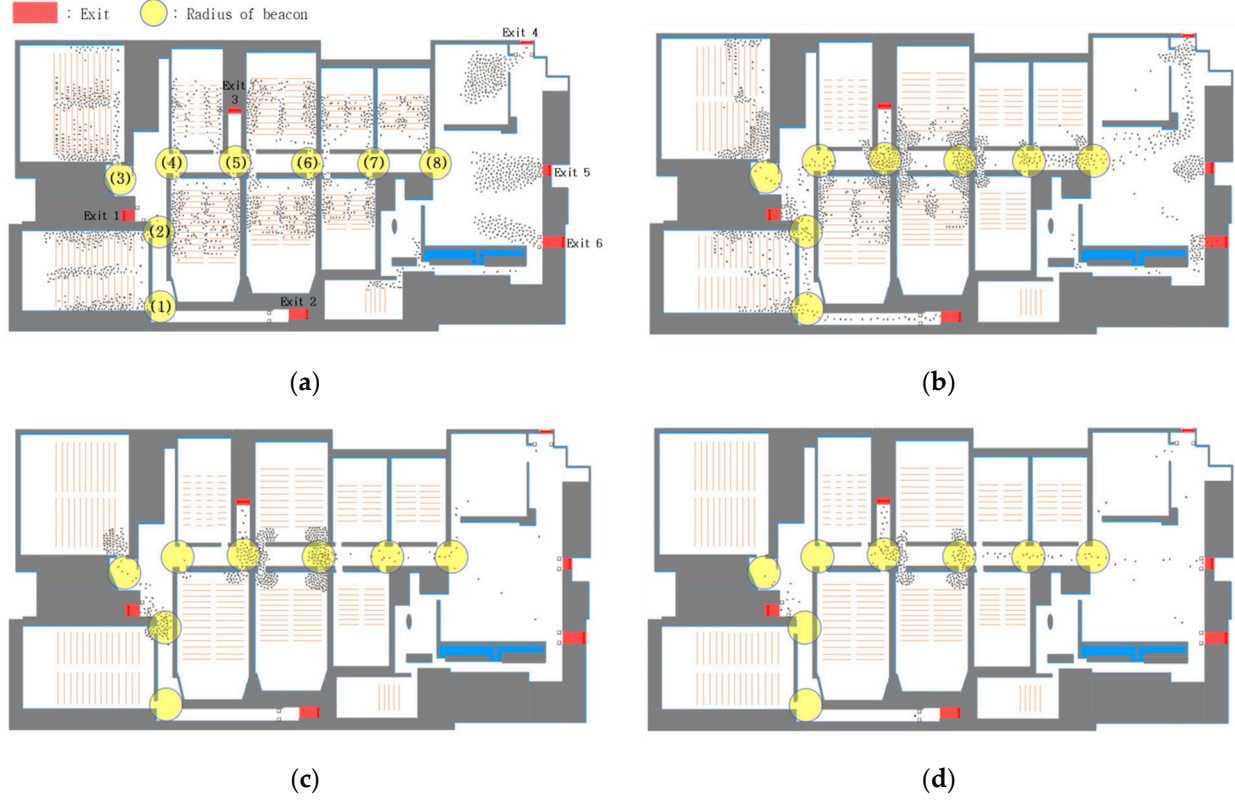

**Figure 9.** Simulation result with the evacuation system: (**a**) 20 s since the evacuation start; (**b**) 60 s; (**c**) 150 s; (**d**) 210 s.

### 3.3. Estimation of the System Effectiveness

Figure 10 shows the total evacuation time result according to RToB and beacon radius parameters. Each point is an average of 15 simulation results. Zero m beacon radius refers to the simulation result when there was no evacuation system. As a result, the evacuation time decreased from 100 to 65 s when the system was introduced, which shows the beacon radius has a significant influence on the decrease of evacuation time. Otherwise, changing the RToB does not significantly affect total evacuation time.

The dispersion of evacuees (population who follows alternative routes) is measured by each RToB and beacon radius range, shown in Figure 11. The result shows that the dispersion increases as the RToB decreases, and beacon radius increases. The maximum number of dispersed evacuees was 458, which accounts for 28.66% of the total evacuees when the RToB and beacon radius parameters were set to 3 s and 5 m, separately.

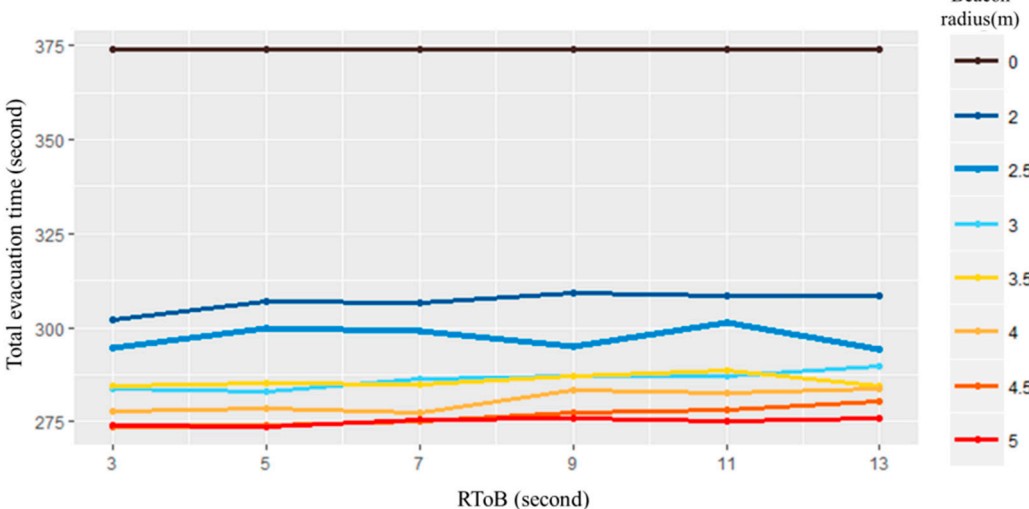

**Figure 10.** Evacuation time as a function of RToB for various beacon radius values.

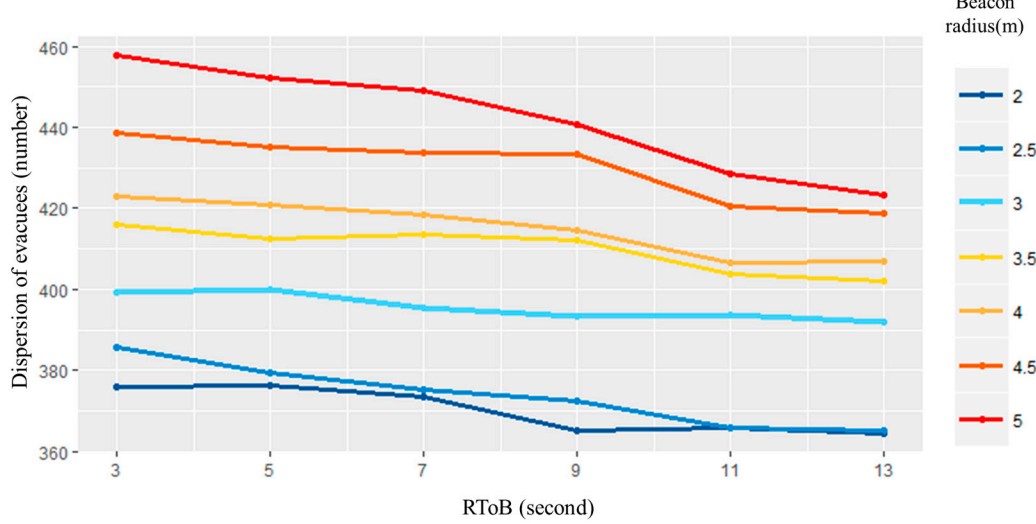

**Figure 11.** Dispersion as a function of RToB for various beacon radius values.

Next, we analyzed the sensitivity of evacuation effectiveness changing the total population in the multiplex and the ratio of the population who complied with detour routes. Sensitivity analysis is often used to simplify a model through factor fixing [39]. However,

we omitted this process since the simulation factors such as shoulder width were validated according to the existing experimental research.

Figure 12A shows the evacuation time decrease ratio by population increase. Over 0% evacuation time decrease ratio indicates the advantage of introducing the system. The maximum evacuation effectiveness appears when the total population is over 70%. On the contrary, total evacuation time increases if there is less than 23% of the population. Figure 12B illustrates dispersed population pattern according to the total population change. The maximum dispersion ratio is around 28% and is observed when the population ratio is greater than 40%. Therefore, the total population must be at least 23% for the system to improve the total evacuation time and 70% to get maximum evacuation effectiveness.

Figure 12C,D shows the sensitivity analysis results according to the compliance ratio of the provided routes. Since the routes are transmitted directly to the individual smart device, people who do not install the application could not comply with the provided routes. Therefore, we need to recommend people to install the application before or when they enter the building to secure greater evacuation effectiveness. The results show the decrease of total evacuation time, and the increase of the size of the dispersed populations as the number of evacuees who follow the detour routes increases.

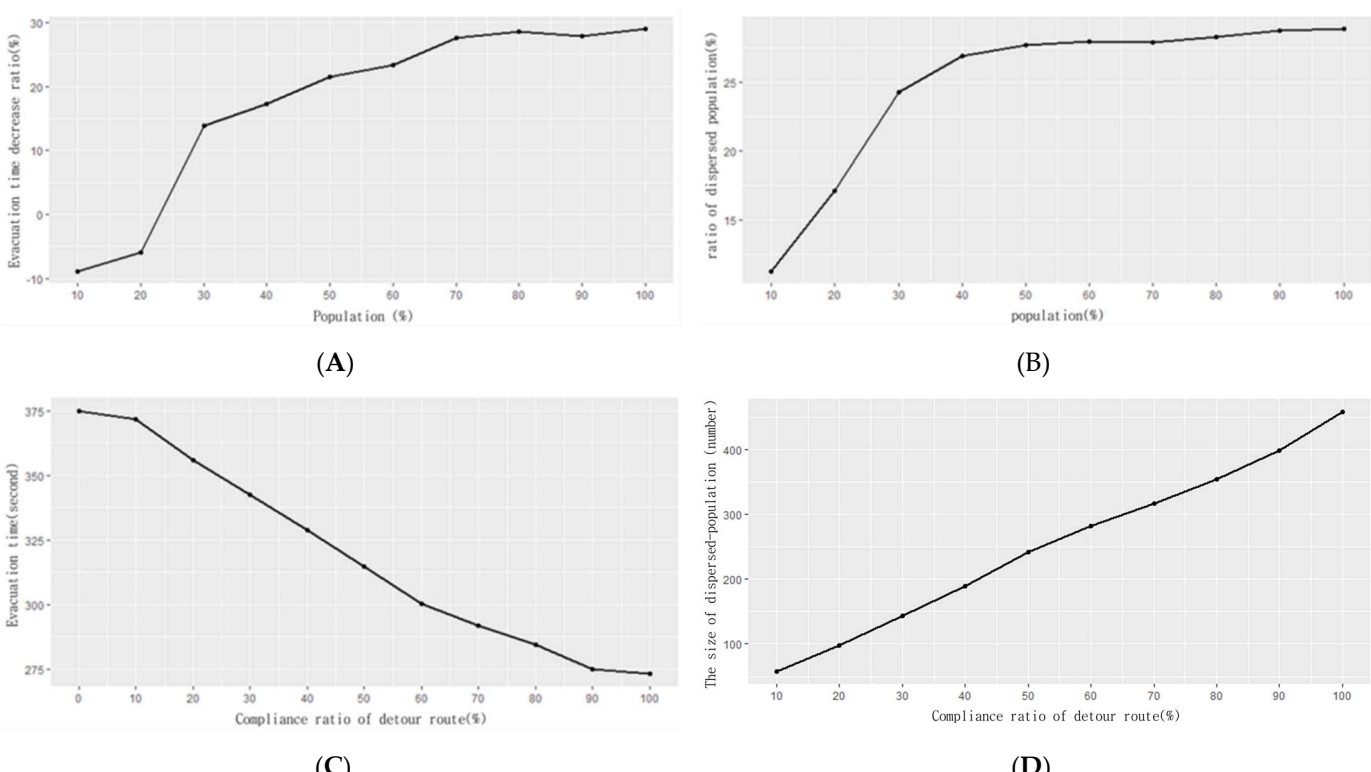

**Figure 12.** Sensitivity analysis result.

## 4. Discussion

To reduce the bottlenecks, much research has studied the correlation between physical indoor structure and human flow, or the creation of a route-finding algorithm in many ways. However, these still might have difficulties in solving the continuous bottlenecks in a complex indoor space.

In this study, we considered two significant problems related to the bottleneck issues we found from the previous evacuation route algorithms. First, serious injuries and casualties could be caused due to the nth bottlenecks that occur when all evacuees are provided the same alternative routes regardless of their locations and rush to the same exit together, which ends up being crowded quickly. Second, evacuation time and collision

accidents increase when the system does not consider dynamic movement so that people may take a long way to another exit despite the fact that bottleneck in front of them could disappear soon. To address these issues, we designed an evacuation system utilizing an IoT sensor machine and beacons. The sensor and beacons have a significant role in distributing people by detecting continuous bottlenecks and providing real time alternative routes.

As a result of ABM simulation applied to this approach, we found that bottlenecks were distributed to each exit and eventually, evacuation time was significantly reduced. This shows that the use of IoT and beacon machines based on LBS provides the theoretical solution for reducing continuous bottlenecks, which existing research has not thoroughly considered. Therefore, we could apply this method to other evacuation system to improve evacuation effectiveness.

Moreover, since there have been many buildings with complex indoor space, it is not easy to establish a customized evacuation plan and optimize evacuation effectiveness. In this research, therefore, we suggested the way to optimize the evacuation system by modulating system parameters so that it can be introduced in another building. First, we estimated how much the RToB and beacon radius affect evacuation efficiency and found optimal parameters for the simulation area. Second, we found that we do not need the evacuation system when the number of people is below 23% since the evacuation time in that case increases instead. These methods can be applied to another building and would contribute to securing maximum evacuation effectiveness when the system is installed. Since this system could be easily developed by attaching various sensors, this can flexibly respond to various situations such as fire, or earthquakes. This will help in reducing danger in a city from either natural or human disaster and contribute to establish a sustainable smart city.

### 5. Conclusions

Bottlenecks during evacuation in a complex building could cause serious injuries or even casualties. Despite much effort to tackle the issues, preventing continuous bottlenecks has not been seriously treated in previous studies. Therefore, we made a prototype evacuation system and validated this using ABM, as well as optimized the system to provide a theoretical solution in any indoor environment.

Through ABM simulation, the system shows a decrease in the continuous occurrence of the bottleneck by letting evacuees who are far from the exit continue to seek alternative routes and those who are already close to the exit keep moving along the original routes. To consider other parameters that could affect the evacuation time and safety, we conducted a sensitivity analysis. Results show the evacuation effectiveness is dependent on the total population and the number of evacuees who comply with the provided routes in the building. Thus, we need to monitor how many people are inside the building to decide to activate this system and encourage people to install the mobile application before they enter, to gain high evacuation effectiveness.

Our research has some advantages that this evacuation system prototype can be installed to any complex shape of a building by using IoT and beacon machines. Moreover, the LBS method for distributing bottlenecks can be easily applied to any other evacuation system to rescue people in a short time. However, this study also has some limitations. For example, in case of terrorism, DoS or DDoS attacks could happen to shut down the system or try to trigger false alarms. To prevent this, the in-line DDoS protection that installs IP protection cloud in our server network to mitigate dirty traffics can be one of the solutions for this system in the future. Second, to make the evacuation simulation environment similar to the real world, only shoulder width and the speed of the evacuees were validated by comparing them to the existing experimental studies. Therefore, the evacuation movement did not consider the sociodemographic characteristics of evacuees or whether they are a group or not. Third, there is a constraint in setting only the single floor as a study area. Given these limitations, future study needs to improve the protection meth-

ods and verify this evacuation system by considering demographic behavioral patterns and multi-layer structures.

**Author Contributions:** Conceptualization and methodology, M.C., S.-J.C. and C.S.H.; IoT machine installation and making the evacuation system, M.C.; establishing ABM simulation, M.C. and S.-J.C.; simulation running and analysis, M.C.; validation of the simulation, M.C. and S.-J.C.; writing—original draft preparation, M.C.; writing—review and editing, M.C., S.-J.C. and C.S.H.; visualization, M.C.; supervision, S.-J.C. and C.S.H. All authors have read and agreed to the published version of the manuscript.

**Funding:** This research received no external funding.

**Institutional Review Board Statement:** Not applicable.

**Informed Consent Statement:** Not applicable.

**Data Availability Statement:** This study published the simulation result dataset. Choi, Moongi (2021), "Density and evacuation Time by RToB and Beacon radius", Mendeley Data, V2, doi:10.17632/8z65y9r35x.2 (https://data.mendeley.com/datasets/8z65y9r35x/draft?a=da9bd35c-4fe8-4732-a922-8b7e9a051462, accessed on 1 July 2021); Choi, Moongi (2021), "Dispersion of evacuees by RToB and Beacon radius", Mendeley Data, V2, doi: 10.17632/h7xjpjftzj.2 (https://data.mendeley.com/datasets/h7xjpjftzj/draft?a=a6943832-899c-4be2-9741-99c0c776ddf5, accessed on 1 July 2021).

**Acknowledgments:** This research was supported by the MSIT(Ministry of Science, ICT), Korea, under the High-Potential Individuals Global Training Program)(2020-0-01593) supervised by the IITP (Institute for Information & Communications Technology Planning & Evaluation).

**Conflicts of Interest:** The authors declare no conflict of interest.

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
