# Peer review of "Relieving Bottlenecks during Evacuations Using IoT Devices and Agent-Based Simulation"

_sustainability, doi:10.3390/su13169465_

Round 1

Reviewer 1 Report

The authors of the article deal with the design of an evacuation system by using IoT sensors, which should streamline the evacuation times of people who are trying to get out of the building as quickly as possible in case of natural disasters, fires, terrorist attacks by using an application they must have pre-installed on their mobile device. This app should show to evacuated people the fastest and the most efficient way to get out of a building in the fastest time possible. This solution was verified by using agent-based simulation. 

Reservations and comments on the proposed evacuation system using IoT devices:

  1. In the article you state that this proposed evacuation system using IoT devices is also suitable for evacuation of persons in case of terrorist attacks. Since it is well known that IoT devices are vulnerable to cyber attacks, it is not mentioned anywhere in the article how you dealt with securing a designed evacuation system against DDoS and DRDoS attacks. Because in the event of a terrorist attack, the attackers first try to shut the evacuation and security systems down, and only after that they will attack the building. And also, when such a system is unsecured, it is very easy to misuse it in order to trigger false alarms.
  2. In case of using the proposed evacuation system, you state that each person has to have an evacuation application pre-installed in their mobile device. Have you also considered that not every person will have a cell phone with them in case of an alarm? What if someone´s battery fails? What if not everyone has a cell phone that is compatible with the proposed application? 
  3. Nowhere in the article is stated whether the given system is already suitable for direct use in practice, or it is only a theoretical basis for other proposed solutions and research. Because in the practice, this does not seem to me to be the best solution, and I find it quite illogical. The reason is that I can not imagine as every visitor who comesto the cinema or to the theater will, before entering the building, install an application in his/her cell phone (if he/she even has it with him/her at all) in the case of an evaluation, as it is mentioned in the article that persons in the case of an evaluation must already have the application installed in the phone.
  4. Will the mentioned simulation times be meaningful even if half of the evaluated people will not have an evacuation application installed? 
  5. At the end of the article, I could not find a „conclusion“ chapter where an overall summary of the proposed solution of advantage/disadvantage could be written in which the proposed solution could be improved in the future.

Author Response

Thank you for reviewing this paper “Relieving bottlenecks during evacuations using IoT devices and agent-based simulation”. We appreciate your detailed review and put our effort to follow. We made a table to investigate your every comment and revise point by point.

Reviewer 2 Report

The paper is an interesting study, looking at ways to tackle issues normally faced during a building evacuation using IoT devices and agent-based simulation to over common bottlenecks. The study is an original work with extensive analyses providing new contributions to the body of knowledge in the associated field, however, the authors should include brief conclusions as part of a new section, which is currently missing from the paper.

Author Response

(The authors gave the same response as above.)

Round 2

Reviewer 1 Report

The authors dealt with all the addressed comments and suggestions regarding the article. The authors have modified and supplemented the missing parts and information, and therefore, I have no further comments or suggestions.